# Ectopic Expression of *AtYUC8* Driven by *GL2* and *TT12* Promoters Affects the Vegetative Growth of Arabidopsis

Chao Tan [1], Jia Song [1], Tan Zhang [1], Mengxiao Liang [1], Suxin Li [1], Huabin Liu [2] and Shuzhen Men [1,*]

1   Tianjin Key Laboratory of Protein Sciences, Department of Plant Biology and Ecology, College of Life Sciences, Nankai University, Tianjin 300071, China; tanchao@mail.nankai.edu.cn (C.T.); j1694134744@163.com (J.S.); zt15291459854@163.com (T.Z.); liang1121232022@163.com (M.L.); 1120180383@mail.nankai.edu.cn (S.L.)
2   College of Life and Health Sciences, Anhui Science and Technology University, Chuzhou 233100, China; liuhuabin2006@126.com
*   Correspondence: shuzhenmen@nankai.edu.cn

**Abstract:** Auxin plays an essential role in regulating Arabidopsis growth and development. *YUCCA* (*YUC*) family genes encode flavin monooxygenases, which are rate-limiting enzymes in the auxin biosynthetic pathway. Previous studies showed that *YUC8* overexpression (*YUC8* OE), as well as ectopic expression of *YUC8* driven by *GL2* (*GLABRA 2*) and *TT12* (*TRANSPARENT TESTA 12*) promoters, which are specifically expressed in the epidermis and inner seed integument, respectively, produced larger seeds compared to the Col. However, the impact of these transgenic lines on the vegetative growth of Arabidopsis remains unclear. Here, we show that the *GL2pro:YUC8-GFP* and *TT12pro:YUC8-GFP* transgenic plants produce a moderate excessive auxin accumulation phenotype compared to the *YUC8* OE. These two transgenic lines produced smaller rosette and leaf, higher plant height, fewer branches, and longer siliques. These data will provide a basis for the study of the relationship between the ectopic expression of auxin synthesis genes and crop yield.

**Keywords:** auxin; *YUCCA*; vegetative growth; yield; ectopic expression; *Arabidopsis thaliana*

## 1. Introduction

As the first plant hormone discovered, auxin plays an important role in all stages of plant growth and development [1,2]. The auxin biosynthesis in plants mainly includes tryptophan (Trp)-dependent and Trp-independent pathways [3–6]. At present, an enormous amount of research focuses on the Trp-dependent pathway, but little is known about the Trp-independent pathway. The mutation of maize tryptophan synthase β leads to the abnormal synthesis of tryptophan but still has a high level of IAA, suggesting the existence of a Trp-independent auxin biosynthesis pathway [7]. Similarly, the Arabidopsis *trp2-1* mutant with a defect in tryptophan synthase β leads to abnormal tryptophan synthesis, in which a higher level of amide- and ester-linked IAA compared with the wild type is still produced [8,9]. The Trp-dependent pathway includes four subpathways that utilize tryptophan metabolism to produce different intermediates, namely, indole-3-acetaldoxime (IAOx), indole-3-acetamide (IAM), tryptamine (TAM), and indole-3-pyruvate (IPA) pathways, respectively [4–6]. Trp synthesized by the shikimic acid pathway in the chloroplasts becomes the central precursor of the four Trp-dependent pathways [6]. The most important enzymes in the IAOx pathway are the cytochrome P450 monooxygenase CYP79B2 and CYP79B3, which catalyze Trp to indole-3-acetaldoxime [10]. Transgenic plants that overexpress the *CYP79B2* gene contain higher levels of free IAA than the wild-type plants and show phenotypes characteristic of auxin overproduction, whereas *cyp79b2 cyp79b3* double mutants accumulate lower levels of auxin and exhibit mild IAA-deficient phenotypes [10]. A recent study has shown that *cyp79b2 cyp79b3* exhibits a phenotype of early aging, revealing its role in age-related developmental processes [11]. Overexpression of the bacterial *iaaM* gene (encodes a Trp

monooxygenase) in plants can produce IAM and lead to excessive-auxin phenotypes, such as epinastic cotyledons, long hypocotyl, short root with long root hairs, and narrower leaves [12]. To identify key genes in the IAM auxin biosynthesis pathway, Gao et al. performed forward genetic experiments in Arabidopsis and identified the *IAM hydrolase 1* (*IAMH1*) and *IAMH2* genes. The *iamh1 iamh2* double mutants are resistant to IAM treatment but display no obvious developmental defects under normal growth conditions, revealing that the IAM pathway plays a weak role in plant auxin synthesis [12]. So far, little is known about the tryptamine auxin biosynthesis pathway. Tryptophan decarboxylase (TDC) is involved in this process, but there is still little research on it [1,13].

The IPA pathway is the main auxin biosynthesis pathway in most plants, which involves two vital reactions. First, Trp is catalyzed to form indole-3-pyruvic acid (IPyA) by the aminotransferase TAA (TRYPTOPHAN AMINOTRANSFERASE OF ARABIDOPSIS), and then indole-3-pyruvic acid undergoes oxidative decarboxylation to form IAA catalyzed by the YUCCA (YUC) flavin monooxygenase [14–21]. Both the TAA and YUC enzymes are encoded by multiple genes in most plants examined. For example, in the Arabidopsis genome, there are five closely related genes encoding the TAA or the TAA-related 1 to 4 (TAR1-4) [14,15]. The single *taa1* or *tar1-4* mutants show no obvious defects, whereas the *taa1 tar1 tar2* triple mutants are lethal at the seedling stage. The *taa1 tar1 tar2* seedlings also bear typical auxin-deficiency phenotypes, such as no primary root and short hypocotyl, with only one cotyledon [18]. YUC is the rate-limiting enzyme of the IPA auxin biosynthesis pathway. Overexpression of the *YUC* gene in plants causes the characteristic phenotype of auxin overproduction [20]. There are a total of 11 *YUC* genes in Arabidopsis, and their expression is strictly regulated with tissue specificity and functional redundancy [20,22,23]. *YUC1*, *YUC2*, *YUC4*, and *YUC6* are mainly involved in the biosynthesis of auxin in shoot apical meristems, leaf primordia, and floral organs, while *YUC3*, *YUC5*, *YUC7*, *YUC8*, and *YUC9* mainly participate in auxin biosynthesis in roots [22–25]. During reproductive growth, *YUC1*, *YUC4*, and *YUC6* are expressed in the ovule integuments and funiculus [26]. *YUC1*, *YUC4*, *YUC8*, *YUC9*, *YUC10*, and *YUC11* are expressed during embryo and seed development [23,24,27]. Consistently, the *yuc3 yuc5 yuc7 yuc8 yuc9* quintuple (*yucQ*) mutants exhibit severe developmental defects in roots and are agravitropic [25]. Ectopic expression of the *YUC3* gene under the control of a shoot-specific promoter does not rescue the abnormal root phenotype of the *yucQ* mutant, suggesting that local auxin synthesis is essential for root development [25]. The *yuc1 yuc2 yuc4 yuc6* quadruple mutants display severe developmental defects in leaf morphology, vasculature tissues, and flower organs [22,23]. The *yuc1 yuc4 yuc10 yuc11* quadruple mutants display severe defects during embryogenesis and produce seedlings without hypocotyl or root [23]. These findings indicate that *YUCs* play an important regulatory role during the development of roots, leaves, floral organs, and embryos in Arabidopsis. In addition, mutations in the *YUC* genes cause the plant to be sensitive to the external environment and reduce the plant's adaptation to abiotic stress [28].

Ectopic expression of auxin-related genes in plants affects plant development. For example, ectopic overexpression of the *AtSAUR41* gene leads to long hypocotyl and root, increased numbers of lateral roots, and twisted stem, whereas its tissue-specific expression driven by the *PIN1* promoter in the stele cells leads to abnormal root meristem with additional cell layers [29]. Ectopic expression of the *OsIAA4* gene in rice reduced the sensitivity to auxin and markedly increased the plant height and tiller angles [30]. *UGT84A2* and *UGT75D1* encode auxin glycosyltransferases, which are specific for indole-3-butyric acid (IBA) [31,32]. *UGT84A2* plays an important role in the *Arabidopsis thaliana* flowering process. The *UGT84A2* overexpression led to the increase in the IBA level in Arabidopsis, which disturbed the dynamic balance of auxin and downregulated the gene expression related to flowering, and the transgenic plants showed the characteristics of late flowering [31]. The ectopic expression of *UGT75D1* in Arabidopsis leads to early seed germination and affects plant growth and development under stress conditions [32]. In addition, the ectopic expression of another IAA-dependent auxin glycosyltransferase UGT74D1 leads to a disordered

distribution of auxin in leaves and petioles and produces leaves with abnormal morphology, leaf vein distribution, and petiole angle [33]. Overexpression of the Arabidopsis *YUC6* gene in sweetpotato can lead to better antioxidant and drought resistance, which is beneficial for their adaptation to environmental stress [34]. The expression pattern of *SAUR19* exhibits asymmetric distribution in the soybean seedling hypocotyl, especially under conditions of gravity and light stimulation. The ectopic expression of *SAUR19* under the control of the 35S promoter can weaken the gravitropic and phototropic responses of soybean seedling hypocotyl [35]. The previous research showed that ectopic expression of the *AtYUC8* gene driven by *GL2* and *TT12* promoters in Arabidopsis produced larger seeds [27]. Consider if these auxin-related genes are overexpressed in crops in a manner of ectopic expression, and it is highly likely to increase crop yield. However, it seems that it is not solely based on the agronomic traits of seed size to determine whether it has the potential to improve agricultural production. Therefore, it is necessary to analyze the vegetative growth of those transgenic plants, such as plant height, the size of leaves, the number of branches, and other morphological characteristics, which may have an important role in crop development. In the previous research, it remains unknown whether the *GL2pro:YUC8* and *TT12pro:YUC8* transgenes affect the vegetative growth of the transgenic plants. Therefore, in this study, the vegetative growth of these transgenic lines was analyzed. The results will provide a basis for the application prospect of these two transgene constructs in agricultural production.

## 2. Materials and Methods

### 2.1. Plant Materials and Growth Conditions

The Arabidopsis transgenic lines used in this article have been described in previous reports: *YUC8* OE [24], *GL2pro:YUC8-GFP* (L6, L8, L10), and *TT12pro:YUC8-GFP* (L8, L11, L12) [27]. The Col-0 (Col) ecotype was used as a wild-type control. Seeds were surface sterilized first in 70% (*v/v*) ethanol (ETOH) for 5 min and then in 1% (*v/v*) Clorox bleach for 10 min, and then were washed five times using sterile ddH$_2$O. After stratification in the refrigerator for 2 to 3 days, the seeds were sown on solid Murashige and Skoog (MS) plates and cultured in a plant incubator at 22 °C under a photoperiod cycle of 16 h light/8 h dark. Seven-day-old seedlings were transplanted to the soil and were grown in a culture room at 22 °C under a photoperiod of 16 h light/8 h dark. The ethanol and Clorox bleach were purchased from Aladdin Reagent Company (Shanghai, China), and the Murashige and Skoog medium powder was purchased from Duchefa Biochemie Company (Haarlem, The Netherlands).

### 2.2. Plant Morphology Measurement

Morphological measurements of Col, *YUC8* OE, *GL2pro:YUC8-GFP* (L6, L8, L10), and *TT12pro:YUC8-GFP* (L8, L11, L12) plants were conducted on rosette diameter, plant height, leaf length/width, petiole length, primary/secondary branch numbers, and stem diameter. The measurement of rosette diameter starts from the third week until the ninth week. The fully expanded fifth rosette leaf of Arabidopsis plants was selected to measure the leaf length, leaf width, and petiole length. Arabidopsis plant height was measured from the fourth week to the ninth week. A vernier caliper was used to measure the diameter of the main stem. The numbers of the primary branches of Arabidopsis are counted from the fourth week to the eighth week, and the secondary branches are from the fifth week to the eighth week. The siliques are collected 14 days after pollination and observed, and photos are taken under a stereo microscope (Leica, Wetzlar, Germany). Image J software 1.4.3.67 (https://imagej.net/ij/ (accessed on 15 May 2023)) is used for measurement.

### 2.3. Statistical Analysis

Statistical differences were analyzed using the one-way ANOVA multiple comparison test. The experimental results are shown as mean ± SD. $p < 0.05$ was determined as statistically significant.

## 3. Results

*3.1. Ectopic Expression of the YUC8 under the Control of GL2 and TT12 Promoters Affects the Size of Arabidopsis Rosette*

The *GL2* and *TT12* are specifically expressed in the epidermis and the innermost layer of ovule and seed integuments of Arabidopsis plants, respectively [36]. We previously obtained the *GL2pro:YUC8-GFP* and *TT12pro:YUC8-GFP* transgenic plants in which *YUC8* was driven by the *GL2* and *TT12* promoters to ectopically express at the epidermis and the inner seed integuments [27]. Here, we analyzed the vegetative growth phenotypes of Col, *YUC8* OE, *GL2pro:YUC8-GFP*, and *TT12pro:YUC8-GFP* transgenic lines. The results showed that the rosette diameter of the *GL2pro:YUC8-GFP* L6 and L8 was significantly smaller than that of the Col, while there was no significant difference between the other transgenic lines and the Col on rosette size in the third week (Figure 1a,b). Starting from the fifth week, the rosette diameter of *TT12pro:YUC8-GFP* L11 and L12 were also significantly smaller than that of the Col (Figure 1b). However, there was no statistically significant difference between *YUC8* OE and Col plants during the entire vegetative growth stage in the rosette diameter (Figure 1b). These results indicate that ectopic expression of *YUC8* affects the size of the rosette of transgenic plants.

*3.2. Ectopic Expression of the YUC8 under the Control of GL2 and TT12 Promoters Affects the Development of Arabidopsis Leaves*

Next, we conducted statistical analysis on the leaf length, leaf width, and petiole length of the fifth leaf of the rosette (Figure 1c). The results showed that there was no significant difference in leaf length between all transgenic plants and Col, except for the *GL2pro:YUC8-GFP* L8 (Figure 1d). The leaf width of *YUC8* OE, all lines of the *GL2pro:YUC8-GFP* and the *TT12pro:YUC8-GFP* L11 was all smaller than that of the Col (Figure 1e). However, there was no significant difference in leaf length/width ratio between *GL2pro:YUC8-GFP* and *TT12pro:YUC8-GFP* transgenic lines and the Col, while *YUC8* OE was significantly higher than the Col (Figure 1g). These results indicate that *YUC8* overexpression leads to narrower leaves. Although similar in morphology to the Col, the *GL2pro:YUC8-GFP* and *TT12pro:YUC8-GFP* transgenic lines produced smaller leaves. Overexpression of the *YUC8* did not affect the growth of leaf petiole (Figure 1f), whereas the petiole length of the *GL2pro:YUC8-GFP* and the *TT12pro:YUC8-GFP* L11 and L12 were significantly shorter compared to that of the Col (Figure 1f). There is no statistically significant difference in the ratio of petiole length to leaf length between Col and all of the transgenic plants (Figure 1h).

*3.3. Ectopic Expression of the YUC8 under the Control of GL2 and TT12 Promoters Affects the Arabidopsis Height but Does Not Affect the Stem Thickness*

Subsequently, we measured the plant height of Col and the transgenic lines from the 4th to the 9th week. The results showed that the plant height of the *YUC8* OE, the L6, and L10 of *GL2pro:YUC8-GFP*, and the L8 and L11 of *TT12pro:YUC8-GFP* were significantly higher than those of the Col, starting from the seventh week (Figure 2a). By the ninth week, the plant height of the *YUC8* OE, all lines of the *GL2pro:YUC8-GFP*, and the L8 and L11 of *TT12pro:YUC8-GFP* were significantly higher compared to the Col (Figure 2a). Then, we measured the diameter of the main stem of these plants. Compared with the Col, the main stems of the *YUC8* OE and the L8 of *TT12pro:YUC8-GFP* plants are thicker, while there was no significant difference between the other lines and the Col (Figure 2b). These results indicate that ectopic expression of the *YUC8* can increase plant height but in general does not affect stem thickness.

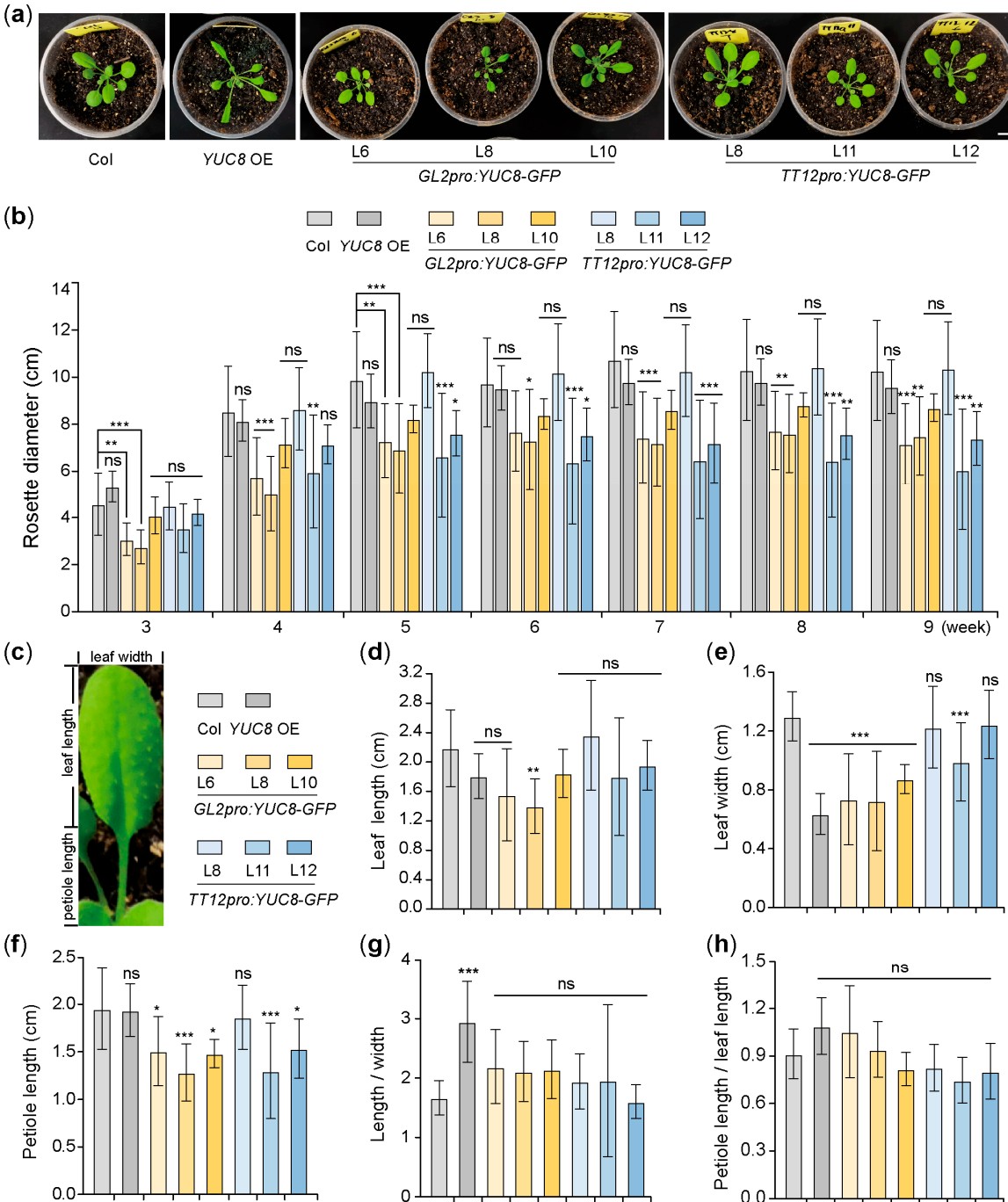

**Figure 1.** Ectopic expression of the *YUC8* gene affects the development of Arabidopsis rosette and leaves. (**a**) Images of three-week-old Col, *YUC8* OE, *GL2pro:YUC8-GFP* (L6, L8, and L10) and *TT12pro:YUC8-GFP* (L8, L11, and L12) transgenic lines; (**b**) The rosette diameter statistics of Col, *YUC8* OE, *GL2pro:YUC8-GFP*, and *TT12pro:YUC8-GFP* transgenic Arabidopsis from 3rd to 9th week; (**c**) Diagram of leaf length, leaf width, and petiole length; (**d**–**h**) Statistics on the leaf length (**d**), leaf width (**e**), petiole length (**f**), leaf length/width (**g**), and petiole length/leaf length (**h**) of the fifth leaf of different transgenic lines and Col Arabidopsis. Error bars represent means ± SD. Significant differences are according to the one-way ANOVA multiple comparison test (*, $p < 0.05$; **, $p < 0.01$; ***, $p < 0.001$; ns = not significant). Bar = 1 cm.

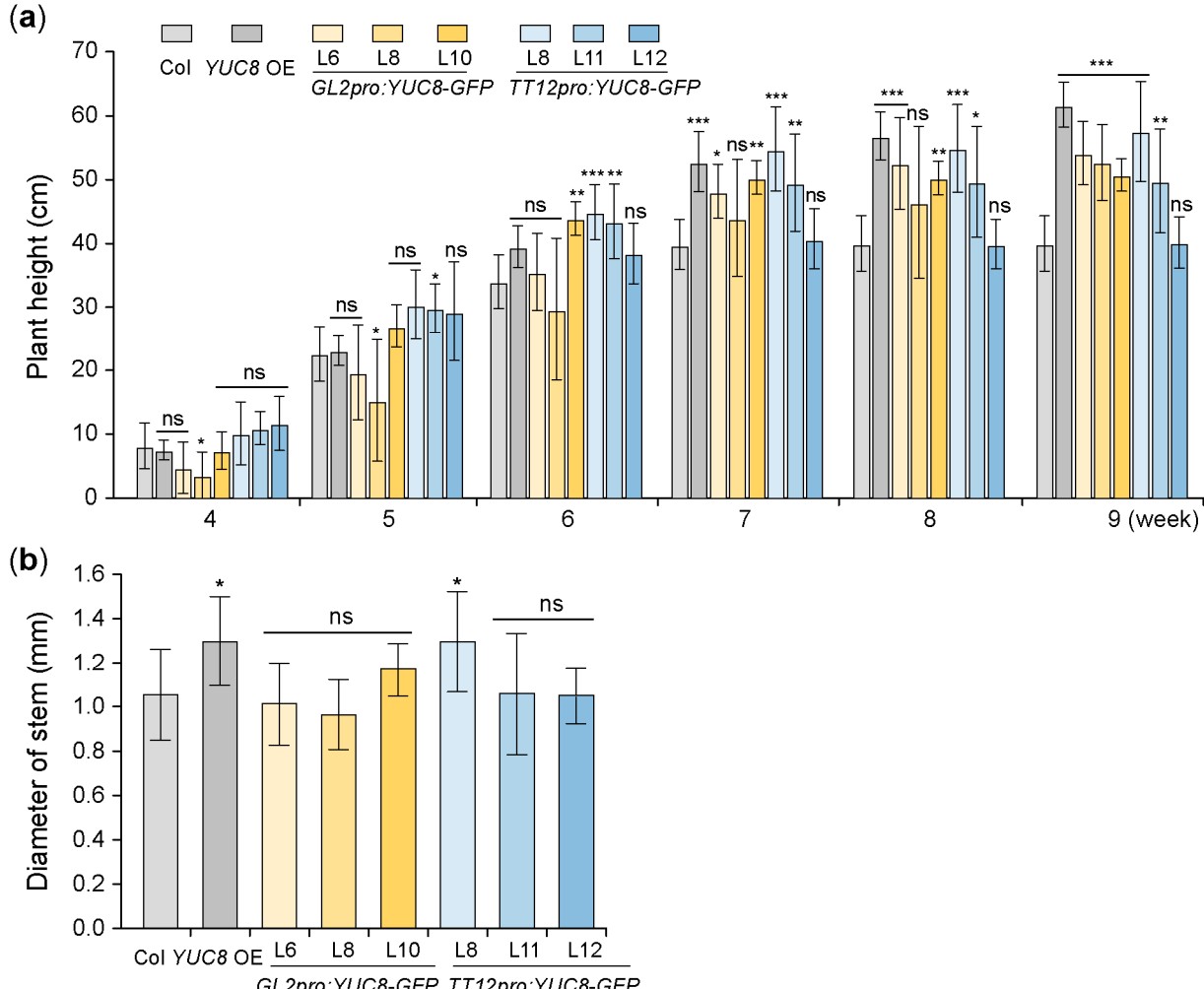

**Figure 2.** Ectopic expression of the *YUC8* gene affects the Arabidopsis height but does not affect the diameter of the stem. (**a**) The plant height statistics of Col, *YUC8* OE, *GL2pro:YUC8-GFP*, and *TT12pro:YUC8-GFP* transgenic Arabidopsis from 4 to 9 weeks. (**b**) The diameter of stem statistics of Col, *YUC8* OE, *GL2pro:YUC8-GFP*, and *TT12pro:YUC8-GFP* transgenic Arabidopsis. Error bars represent means ± SD. Significant differences are according to the one-way ANOVA multiple comparison test (*, $p < 0.05$; **, $p < 0.01$; ***, $p < 0.001$; ns = not significant).

*3.4. Ectopic Expression of the YUC8 under the Control of GL2 and TT12 Promoters Inhibits the Branching of Arabidopsis Plants*

Finally, we analyzed the numbers of the primary and secondary branches of these transgenic lines. Starting from the fifth week, the primary branch numbers of the *YUC8* OE, *GL2pro:YUC8-GFP*, and *TT12pro:YUC8-GFP* lines were significantly lower than that of the Col (Figure 3a). Starting from the sixth week, the secondary branch numbers of the *YUC8* OE, *GL2pro:YUC8-GFP*, and *TT12pro:YUC8-GFP* lines were significantly lower than that of the Col (Figure 3b). These results indicate that ectopic expression of *YUC8* inhibits the branching of Arabidopsis plants and therefore enhances apical dominance.

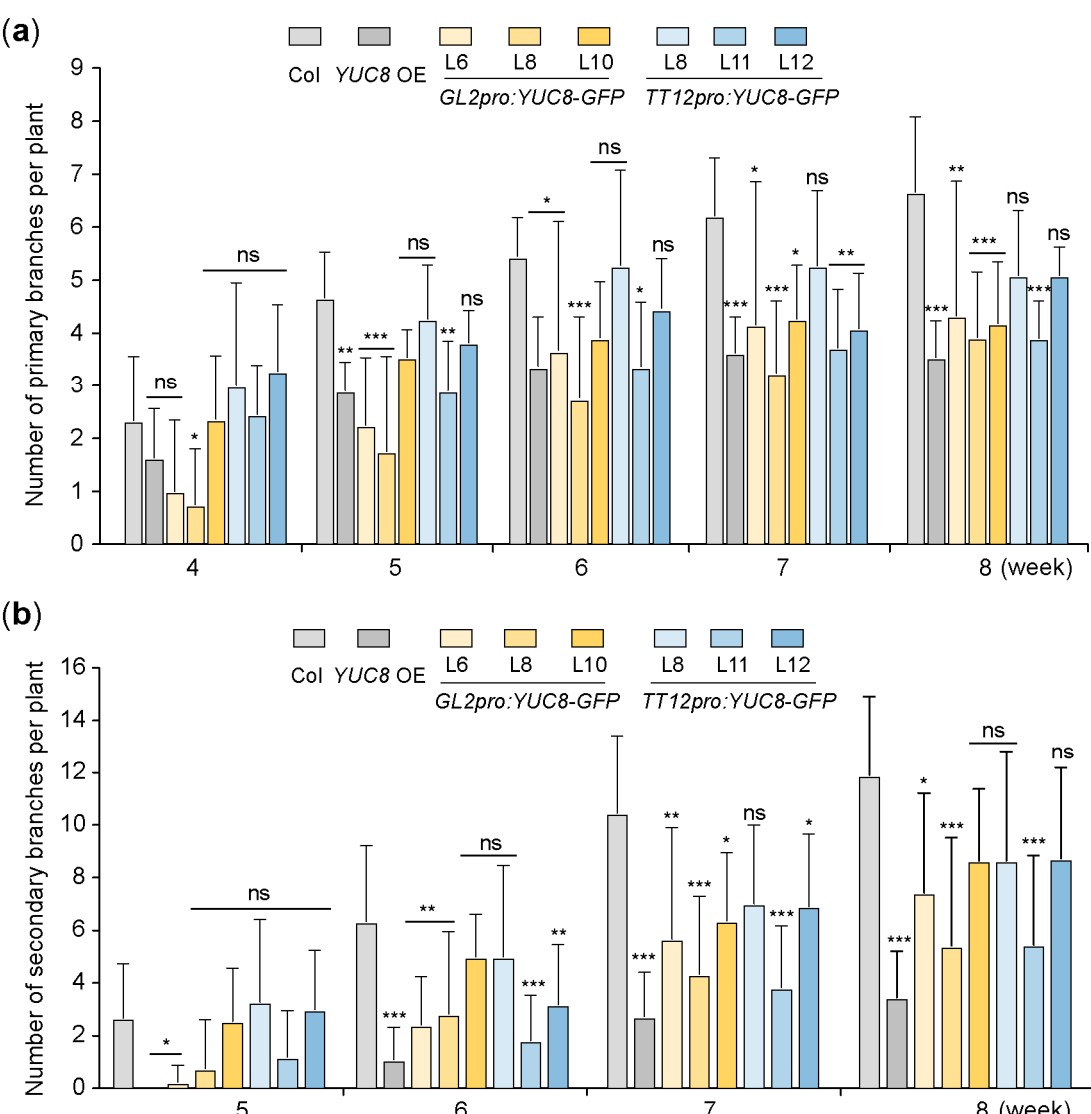

**Figure 3.** Ectopic expression of the *YUC8* gene affects the development of Arabidopsis branches. (**a**) The primary branch statistics of Col, *YUC8* OE, *GL2pro:YUC8-GFP*, and *TT12pro:YUC8-GFP* transgenic Arabidopsis from 4 to 8 weeks. (**b**) The secondary branch statistics of Col, *YUC8* OE, *GL2pro:YUC8-GFP*, and *TT12pro:YUC8-GFP* transgenic Arabidopsis from 5 to 8 weeks. Error bars represent means ± SD. Significant differences are according to the one-way ANOVA multiple comparison test (*, $p < 0.05$; **, $p < 0.01$; ***, $p < 0.001$; ns = not significant).

### 3.5. Ectopic Expression of the YUC8 under the Control of GL2 and TT12 Produce Longer Siliques

The previous research showed that ectopic expression of the *YUC8* gene under the control of the *GL2* and *TT12* promoters in Arabidopsis produced larger seeds [27]. We further analyzed the relative silique length of the *GL2pro:YUC8-GFP* and *TT12pro:YUC8-GFP* transgenic plants 14 days after pollination. The results showed that the silique length of the *YUC8* OE and the *TT12pro:YUC8-GFP* L11 and L12 was not significantly different compared to that of the Col. However, the relative silique length of the L6, L8, and L10 of the *GL2pro:YUC8-GFP* and the L8 of the *TT12pro:YUC8-GFP* transgenic plants was significantly longer than those of the Col (Figure 4). These results indicate that ectopic expression of the *YUC8* can produce longer siliques.

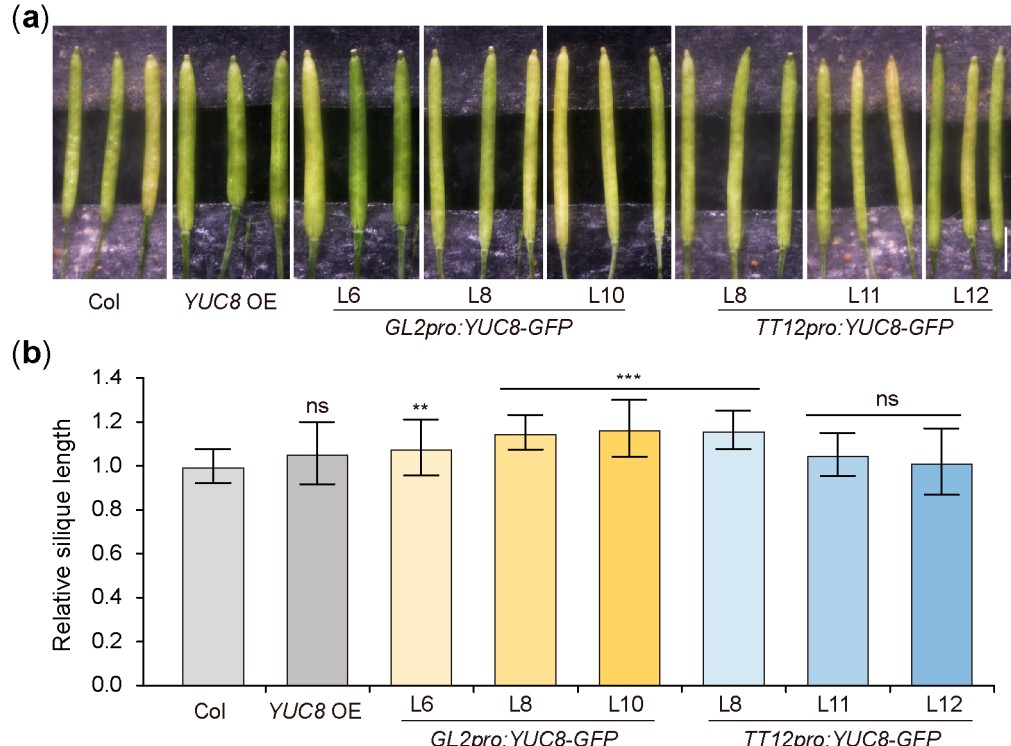

**Figure 4.** Ectopic expression of the *YUC8* gene produces longer siliques. (**a**) The siliques 14 days post pollination of Col, *YUC8* OE, *GL2pro:YUC8-GFP*, and *TT12pro:YUC8-GFP* transgenic Arabidopsis. (**b**) The relative silique length of Col, *YUC8* OE, *GL2pro:YUC8-GFP*, and *TT12pro:YUC8-GFP* transgenic Arabidopsis. Error bars represent means ± SD. Significant differences are according to the one-way ANOVA multiple comparison test (**, $p < 0.01$; ***, $p < 0.001$; ns = not significant). Bar = 0.5 mm.

## 4. Discussion

Gene ectopic expression contributes to its functional analysis. The *JAZ* gene encodes a transcription inhibitor in the jasmonic acid (MeJA) signal transduction pathway. In order to explore the function of *JAZ* in rice, Sun et al. expressed *OsJAZ* under the control of the 35S promoter in Arabidopsis, verified the stability of *OsJAZ* through molecular technology, and found that ectopically expressed *JAZ* led to reduced disease resistance of Arabidopsis and abnormal root and flower organs development [37]. The *PpeDAM6* gene exhibits a high expression level in the peach leaf and bud and mainly regulates the release of bud dormancy. Ectopic expression of the *PpeDAM6* gene in European plums alters its vegetative growth and delays its development with abnormal internodes elongation and disturbances hormone regulatory network [38]. *MYB5* is an *R2R3 MYB* transcription factor involved in the synthesis of secondary cell walls in fruit trees. Chen et al. ectopically expressed the *MdMYB5* gene in Arabidopsis to investigate the mechanism by which *MYB5* affects apple fruit development. The transgenic plants exhibited a dwarf phenotype with thickened secondary cell walls caused by the increase in lignin and cellulose content. Further research showed that *MdMYB5* could regulate the formation of apple secondary cell walls by regulating the expression of cellulose and lignin biosynthesis-related genes, laying the foundation for the study of the regulatory mechanism of *MdMYB5* in apples [39]. The *IQD* genes play an important role in plant response to abiotic stress. *BrIQD35* is mainly expressed in the root, stem, leaf, and other vegetative organs of Chinese cabbage, which can effectively cope with drought and salt stress on plants [40]. Ectopic expression of *BrIQD35* in *Nicotiana benthamiana* showed no significant change in salt tolerance, but significantly increased drought resistance [40]. Phosphorelay by a two-component system (TCS) is a signal transduction mechanism involved in plant response to external stimuli, which is conserved in evolution. *GmHP08* is one of the two-component system members in soybean

and participates in the regulation of drought tolerance. Compared with the wild-type, ectopic expression of *GmHP08* in Arabidopsis exhibited better water storage capacity and a more sensitive response to ABA under water deficiency conditions. Further studies showed that the *GmHP08* gene can mediate plant drought resistance by regulating the expression of ABA synthesis and signaling genes [41]. The *GS5* gene encodes a serine carboxypeptidase-like protein (SCPL) in rice, and can positively regulate the size and weight of rice seeds, which has the potential for improving rice yield and quality. The ectopic expression of the *GS5* gene under the control of the *ZmMRP-1* promoter, which is specifically expressed in the endosperm, exhibits significant improvement in grain filling rate and increased starch particle size and grain weight [42].

The previous study has shown that the *yuc8* mutant produced smaller seeds, while ectopic expression of *YUC8* driven by the *GL2* and *TT12* promoters produced larger seeds [27]. It may provide a strategy for increasing grain yield in crops. Since crop yield is not only contributed by grain size but also by traits such as branch numbers, plant size, and plant architecture [43–45], the effects of *GL2pro:YUC8-GFP* and *TT12pro:YUC8-GFP* on the vegetative growth of Arabidopsis was further investigated in this study. The results showed that ectopic expression of *YUC8* led to higher plants with longer siliques, smaller leaves, and fewer branches, whereas their stems were not thicker than that of the wild type. This may affect the lodging resistance of the plant and be detrimental to yield. The longer siliques induced by ectopic expression of *YUC8* seem to have certain beneficial values in agricultural production, which means that the number and size of seeds wrapped in the pericarp may increase, or simply the increase in pericarp cell numbers or the expansion of pericarp cells.

Another interesting issue in this study is that ectopic expression of *YUC8* in the endothelium layer of the integument can cause a phenotype of moderate excessive auxin accumulation during vegetative growth. Auxin plays an essential regulating role in plant morphogenesis, root development, male and female gametophyte development, and embryogenesis [6,22,23,46]. Auxin exerts its regulatory function by establishing maximum or minimum accumulation in tissues or organs. The local auxin biosynthesis, polar auxin transport, and signal transduction jointly maintain auxin homeostasis in vivo through the complex gene regulatory network. This notion is supported by numerous pieces of research. For example, the *wei8 eir1* (*wei8* is a mutant allele of the *TAA1* gene, and *eir1* is a mutant allele of the *PIN2* gene) and *wei8 aux1* double mutants show additive defects in the maintenance of the root meristem [47]. Overproduction of auxin in the shoot cannot rescue the root developmental defect of the *yucQ* auxin biosynthesis mutant [25]. The auxin-resistant mutant *axr1-3* can suppress the auxin overproduction phenotype induced by the overexpression of the *iaaM* gene in Arabidopsis [48]. It is so far not clear how the auxin synthesized in the inner integument affects the overall vegetative growth of the plant. One plausible explanation is that the auxin efflux proteins PIN1 or PIN3 located in the funiculus might mediate the auxin flow between the integument and the maternal tissue. It has been reported that auxin transport proteins play an important role in mediating auxin signal communication and dynamic distribution between reproductive organs and maternal tissues [26,49,50]. Previous research has found that PIN1, PIN3, and PIN6 are located in the funiculus and might mediate auxin flow between the ovule and the maternal tissue, which may play an important role in the synergistic development of anthers and ovules [26]. Meanwhile, previous research also found that PIN3 mediates the dynamic distribution of auxin in the seed coat, thereby regulating seed development [27]. In summary, our study provides some new insights into the application of tissue-specific expression of auxin biosynthesis genes in agricultural productivity.

## 5. Conclusions

Overexpression of auxin synthesis genes such as *YUCs* can affect plant morphogenesis in *Arabidopsis thaliana*. In this study, we employ the *GL2* and *TT12* promoters which are specifically expressed in the epidermis and inner seed integuments to drive the ectopic

expression of *YUC8* in Arabidopsis, resulting in a moderate excessive auxin accumulation phenotype. Ectopic expression of *YUC8* reduces the size of the rosette and leaves in Arabidopsis but does not affect the morphology of the leaves. Meanwhile, ectopic expression of *YUC8* leads to higher plant height and fewer primary and secondary branches. Ectopic expression of the *YUC8* gene also produces longer silique. Our research may have two limitations. Firstly, we did not detect the levels of auxin in the vegetative organs, such as roots, leaves, and stems by employing auxin response maker *DR5rev:GFP* or *DR5:GUS*. This also makes it unclear how auxin synthesized by the integument affects plant development. Secondly, the impacts of these two constructs on other crops need to be analyzed to determine whether they can truly be applied to practical productivity. However, it is undeniable that compared to *YUC8* overexpression under the control of the 35S promoter, ectopic expression of the *YUC8* gene under the control of the *GL2* and *TT12* promoters produces a phenotype of moderate excessive auxin accumulation, which to a considerable extent alleviates the negative effects of excessive auxin on plants. All in all, our research may provide a basis for further research on the ectopic expression of auxin-related genes in agricultural productivity.

**Author Contributions:** S.M. conceived the project and designed the experiments. C.T. performed the experiments. J.S., T.Z., S.L. and M.L. helped with the experiments. S.M. and C.T. analyzed the data and wrote the manuscript, with contributions from H.L. All authors have read and agreed to the published version of the manuscript.

**Funding:** This work was funded by the National Natural Science Foundation of China (32070281, 31870230, and 91417308 to S.M.).

**Institutional Review Board Statement:** Not applicable.

**Informed Consent Statement:** Not applicable.

**Data Availability Statement:** All relevant data can be found in the manuscript.

**Acknowledgments:** We thank the Arabidopsis Biological Resource Center (ABRC) for providing mutant seeds, and John Innes Centre for providing the pGREENII0229 vector.

**Conflicts of Interest:** The authors declare there is no conflict of interest. The funders had no role in the design of the study; in the collection, analyses, or interpretation of data; in the writing of the manuscript, or in the decision to publish the results.

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
