# Peer review of "Ectopic Expression of AtYUC8 Driven by GL2 and TT12 Promoters Affects the Vegetative Growth of Arabidopsis"

_2674-1024, doi:10.3390/seeds2030021_

Round 1
Reviewer 1 Report
In the current manuscript, Song et al. show that the ProGL2:YUC8-GFP and ProTT12:YUC8-GFP transgenic plants produce a moderate excessive auxin accumulation phenotype compared to the YUC8 OE. These two transgenic lines produced smaller rosette and leaf, higher plant height, and fewer branches. These data will provide a basis for the study of the relationship between the ectopic expression of auxin synthesis genes and the breeding of leafy vegetable crops. Although the topic is attractive, there are some concerns that should be addressed.
-Generally, the manuscript is well organized but has some typographical and grammatical errors.
-L113-116. The authors should clarify the hypothesis and objective of this study.
-The results obtained in this study are interesting. The discussion is presented correctly. However, the discussion should be improved, and previous studies should be discussed more.
- Material and research methods are presented appropriately. The experimental setup and the description in the methods section are well structured, and the statistical analysis is correctly performed.
- I suggest that the authors mention the limitations of the present study in the conclusion part and specify the follow-up tests.
Author Response
- L113-116. The authors should clarify the hypothesis and objective of this study.
Reply: According to the comments of the reviewers, we have made revisions to the introduction of the article and added the objective of the research.
- The results obtained in this study are interesting. The discussion is presented correctly. However, the discussion should be improved, and previous studies should be discussed more.
Reply: According to the comments of the reviewers, we have improved the discussion and added the previous studies in discussion section.
- Material and research methods are presented appropriately. The experimental setup and the description in the methods section are well structured, and the statistical analysis is correctly performed.
Reply: Thank you for the reviewer's affirmation.
- I suggest that the authors mention the limitations of the present study in the conclusion part and specify the follow-up tests.
Reply: According to the comments of the reviewers, we modify the conclusion.
Reviewer 2 Report
Manuscript ID: seeds-2446435
In this manuscript, the authors reported interesting observations of vegetative growth phenotypes of transgenic Arabidopsis lines carrying AtYUC8 gene driven by either GL2 or TT12 promoters. These lines, which have previously been characterized by this team with a focus on seed development (Liu et al., Plant Journal, 2023), are being examined in this manuscript for variations in vegetative growth compared to the wild type. This encompasses traits such as rosette size, plant height, stem diameter, and branch number. The authors attempt to correlate these observed differences to YUC8's established role in auxin biosynthesis. However, additional evidence is required to confirm such a direct link, and I have several concerns about the current manuscript:
1. Upon reviewing the Liu et al. 2023 paper, I could only locate the expression pattern of these two transgenic lines (ProGL2:YUC8-GFP and ProTT12:YUC8-GFP) by GFP signals during seed development. As this study focuses on vegetative growth phenotypes, could the authors provide any information on possible GFP signals in the vegetative tissues, such as the leaves, stems, etc.?
2. Given that the GL2 and TT12 promoters are known to be specific to outer and inner integument seed coat tissues, respectively (as stated by the authors and previous literature), isn’t it surprising if these lines displayed significant phenotypic differences compared to the wild type, even in the absence of YUC8-GFP expression in the leaves, stem, and branches?
3. The study seems to lack direct measurement of auxin levels or an indirect auxin reporter (for example, the DR5 lines used in Liu et al. 2023).
4. I noticed a lack of an appropriate negative control. The Col wild type serves as the comparative standard, and a better negative control would be lines with ProGL2-GFP and ProTT12-GFP that lack the YUC8 gene, wouldn't you agree? While I understand the impracticality of generating these transgenic lines just for revision of this manuscript, I wanted to highlight this consideration for your attention.
5. While the introduction and discussion sections are comprehensive, some explanations of plant hormones (particularly auxin) and YUCCA proteins do not seem directly relevant to this manuscript.
In conclusion, I believe it would be more productive to focus on your findings and observations, rather than making very strong claims. As stated earlier, I'm not suggesting an extensive series of additional experiments. Rather, I encourage you to interpret your observations within their limitations, point out potential future directions, and redraft the discussion (and introduction) with greater care.
The language is ok, but can be improved.
Author Response
- Upon reviewing the Liu et al. 2023 paper, I could only locate the expression pattern of these two transgenic lines (ProGL2:YUC8-GFP and ProTT12:YUC8-GFP) by GFP signals during seed development. As this study focuses on vegetative growth phenotypes, could the authors provide any information on possible GFP signals in the vegetative tissues, such as the leaves, stems, etc.?
Reply: This is a great question. Firstly, in previous studies, we determined that these two constructs were able to correctly express the fusion protein through tobacco transient expression. Secondly, since our previous research mainly focused on seed development, GFP signals can be observed in the seed coat. The GL2 gene is expressed in all epidermal cells of plants, so we observed the ProGL2:YUC8-GFP signal in the roots, and we did detected the GFP signal in roots (data not shown). However, TT12 is only expressed in the integument of the ovule and seed, which has been reported in previous literature. Therefore, we have not observed GFP signals in vegetative tissues of ProTT12:YUC8-GFP transgenic plants. But, we will further observe the expression of these two constructs in vegetative tissues in future research.
- Given that the GL2 and TT12 promoters are known to be specific to outer and inner integument seed coat tissues, respectively (as stated by the authors and previous literature), isn’t it surprising if these lines displayed significant phenotypic differences compared to the wild type, even in the absence of YUC8-GFP expression in the leaves, stem, and branches?
Reply: We have also focused on this interesting phenomenon, so we came up with some ideas in the discussion section of the modified version. One possible reason is that the polar localization of auxin efflux proteins PIN1 or PIN3 located in the funicle mediates the auxin flow between the integument and the maternal tissue. The auxin transport proteins indeed play an important role in mediating auxin signal communication and dynamic distribution between reproductive organs and maternal tissues. Research has found that PIN1, PIN3, and PIN6 located in the funicle mediate auxin flow between the ovule and the maternal tissue, which may play an important role in the synergistic development of anthers and ovules. Meanwhile, our previous research also found that PIN3 mediates the dynamic distribution of auxin in the seed coat, thereby regulating embryonic development. In fact, we also find out some valuable information from our data. There is no significant difference in plant height and rosette between ProTT12: YUC8-GFP transgenetic plants and the Col in four-week-old Arabidopsis thaliana, while the overexpression of YUC8 driven by GL2 promoter has a significant effect on the early development of Arabidopsis thaliana compared to the Col. For example, ProGL2:YUC8-GFP transgenic plants have a small rosette. Starting from the fifth week, when Arabidopsis begins to bloom, ectopic expression of YUC8 driven by TT12 promoter excessively synthesis auxin in the seed coat or ovule integument, which may be transported to the maternal tissue through the funicle. These are only conclusions based on existing experimental data, and we will verify them in future experiments.
- The study seems to lack direct measurement of auxin levels or an indirect auxin reporter (for example, the DR5 lines used in Liu et al. 2023).
Reply: Indeed, we did not pay attention to auxin levels in ProGL2:YUC8-GFP and ProTT12:YUC8-GFP plants by employing DR5:GFP. We will further conduct this assay in future research.
- I noticed a lack of an appropriate negative control. The Col wild type serves as the comparative standard, and a better negative control would be lines with ProGL2-GFP and ProTT12-GFP that lack the YUC8 gene, wouldn't you agree? While I understand the impracticality of generating these transgenic lines just for revision of this manuscript, I wanted to highlight this consideration for your attention.
Reply: Thank you for the reviewer's suggestions.
- While the introduction and discussion sections are comprehensive, some explanations of plant hormones (particularly auxin) and YUCCA proteins do not seem directly relevant to this manuscript.
Reply: According to the comments of the reviewers, we modify the introduction and discussion sections.
Reviewer 3 Report
The manuscript entitled ”Ectopic expression of AtYUC8 driven by GL2 and TT12 promoters affects the vegetative growth of Arabidopsis” is interesting and acceptable after some revision.
The authors used the ProGL2:YUC8-GFP and ProTT12:YUC8-GFP for the analysis, so the localization of gene expression in vegetative organs can be analyzed as same as the authors published for seeds previously in the Plant Journal 2013. If the authors have the expression analysis in vegetative organs, please add them to this report.
It is important to discuss the relationship between auxin concentrations and distributions in each part and the morphological changes by related gene regulation. Please add a more concentrated discussion on the function of IAA.
Author Response
The authors used the ProGL2:YUC8-GFP and ProTT12:YUC8-GFP for the analysis, so the localization of gene expression in vegetative organs can be analyzed as same as the authors published for seeds previously in the Plant Journal 2013. If the authors have the expression analysis in vegetative organs, please add them to this report.
Reply: This is a great suggestion. But unfortunately, we did not conduct expression analysis in vegetative organs. I can provide some explanations. Since our previous research mainly focused on seed development, GFP signals can be observed in the seed coat. The GL2 gene is expressed in all epidermal cells of plants, so we observed the ProGL2:YUC8-GFP signal in the roots, and we did detected the GFP signal in roots (data not shown). However, TT12 is only expressed in the integument of the ovule and seed, which has been reported in previous literature. Therefore, we have not observed GFP signals in vegetative tissues of ProTT12:YUC8-GFP transgenic plants. But, we will further detect the expression of these two constructs in vegetative tissues in future research.
It is important to discuss the relationship between auxin concentrations and distributions in each part and the morphological changes by related gene regulation. Please add a more concentrated discussion on the function of IAA.
Reply: According to the comments of the reviewers, we modify the discussion section.
Reviewer 4 Report
The manuscript titled " Ectopic expression of AtYUC8 driven by GL2 and TT12 promoters affects the vegetative growth of Arabidopsis" aims to understand the effects of ectopic expression of the AtYUC8 gene driven by GL2 and TT12 promoters in Arabidopsis seed coats on vegetative growth of the transgenic plants (ProGL2:YUC8 and ProTT12:YUC8). The authors concluded that the plant morphogenesis in Arabidopsis thaliana can be influenced by the overexpression of genes involved in auxin synthesis, such as YUC. As a result, a phenotype of moderate excessive auxin accumulation was observed. The ectopic expression of YUC8 led to a reduction in the size of the rosette and leaves in Arabidopsis, while the leaf morphology remained unaffected. Additionally, the ectopic expression of YUC8 resulted in increased plant height and a decrease in the number of primary and secondary branches. These findings could contribute to our understanding of the ectopic expression of auxin-related genes and their potential implications in agricultural production. This research could open an avenue for future studies on this topic. While the topic is of significant relevance and general interest to the journal's readership, several concerns need to be addressed before publication.
· The authors are highly recommended to avoid using a personal pronoun (e.g., We, our, etc.); they can use the third party in the past tense's passive voice.
· The authors are strongly advised to carefully review the manuscript to address grammar and other editing issues.
· To ensure reader comprehension, it is essential to provide the full name associated with any abbreviation at its first mention in the manuscript. This practice enables readers who may not be familiar with the abbreviated terminology to follow along and understand the content.
· In the Material and Methods section, it is important to include or complete the sources of all chemicals, software, and equipment by adding the city, state, and country information. This additional detail provides readers with specific information about where these items were sourced, ensuring transparency, and facilitating reproducibility.
· Lines 84-85, please use one style to write the gene name either large letters or small letters throughout the manuscript.
· Some statements in the discussion section need to be supported with the appropriate citation.
Minor editing of English language required!
Author Response
- The authors are highly recommended to avoid using a personal pronoun (e.g., We, our, etc.); they can use the third party in the past tense's passive voice.
Reply: Thank you for the reviewer's suggestions.
- The authors are strongly advised to carefully review the manuscript to address grammar and other editing issues.
Reply: Thank you for the reviewer's suggestions.
- To ensure reader comprehension, it is essential to provide the full name associated with any abbreviation at its first mention in the manuscript. This practice enables readers who may not be familiar with the abbreviated terminology to follow along and understand the content.
Reply: Thank you for the reviewer's suggestions. We have modified it.
- In the Material and Methods section, it is important to include or complete the sources of all chemicals, software, and equipment by adding the city, state, and country information. This additional detail provides readers with specific information about where these items were sourced, ensuring transparency, and facilitating reproducibility.
Reply: Thank you for the reviewer's suggestions. We have modified it.
- Lines 84-85, please use one style to write the gene name either large letters or small letters throughout the manuscript.
Reply: Thank you for the reviewer's suggestions. We have modified it.
- Some statements in the discussion section need to be supported with the appropriate citation.
Reply: Thank you for the reviewer's suggestions. we modify the discussion section.
Round 2
Reviewer 1 Report
All the comments have been addressed.
Author Response
We are very grateful for the reviewer's help.
Reviewer 2 Report
The authors have addressed my initial inquiries and concerns, refining the manuscript accordingly in this second version of manuscript. However, the introduction, discussion, and conclusion sections still contain unnecessary content. Essentially, the authors are presenting unanticipated vegetative growth characteristics from transgenic lines previously developed in their laboratory, which were previously reported to influence seed development. I recommend (once again) that the authors remove unnecessary content from the introduction, and in the discussion section, focus on the observed phenotypes. I asked about why a seed coat-specific promoter would affect vegetative plant growth, because this is the gap between your expected phenotype and previously reported results. The authors responded to this by mentioning future research plans and some unreported data (for instance, the GFP signals in vegetative tissues), which I understand and I am not expecting the authors to provide additional data. Nonetheless, such considerations should be integrated into your discussion section, rather than making indeterminate statements. Last but not least, the authors should try to avoid using informal English language, such as phrases like "we should" or "we need to".
Could be improved.
Author Response
However, the introduction, discussion, and conclusion sections still contain unnecessary content.
Response: We have deleted some unnecessary content. Such as the second sentence in the “Introduction” part: The discovery of auxin de novo synthesis is inseparable from the development of genetics and biochemistry [2]. We also deleted the redundant sentences about the YUC genes in introduction part and the redundant sentences about iaam, yucca, YUC8 and YUC9 etc. in the discussion part.
I asked about why a seed coat-specific promoter would affect vegetative plant growth, because this is the gap between your expected phenotype and previously reported results.
Response: in this version of the manuscript, we discussed this in the last paragraph of the discussion part.
The authors should try to avoid using informal English language, such as phrases like "we should" or "we need to".
Response: We have revised these sentences. For example, In the last paragraph of the Introduction, the “We also need to analyze…..” was changed to “Therefore, it is necessary to analyze…….”.
Reviewer 3 Report
The manuscript has been revised well, and I agree to accept it for publication.
Author Response

(The authors gave the same response as above.)
